# Influence of Orographic Factors on the Distribution of Lichens in the Franz Josef Land Archipelago

**DOI:** 10.3390/plants13020193

**Published:** 2024-01-10

**Authors:** Sergey Kholod, Liudmila Konoreva, Sergey Chesnokov

**Affiliations:** 1Department of Botany, St. Petersburg State University, Universitetskaya Emb., 7–9, St. Petersburg 199034, Russia; sergeikholod@yandex.ru (S.K.); ajdarzapov@yandex.ru (L.K.); 2Komarov Botanical Institute of the Russian Academy of Sciences, Professor Popov St. 2, St. Petersburg 197022, Russia; 3Avrorin Polar-Alpine Botanical Garden-Institute of Kola Scientific Centre of the Russian Academy of Sciences, St. Botanical Garden, 8, Kirovsk 184256, Russia

**Keywords:** lichens, altitude, distance to glacier, slope steepness, exposure, number of species, cover, multiple regression analysis, PCA

## Abstract

During a geobotanical study of the Franz Josef Land archipelago, 111 lichen species were recorded on 130 sample plots. The significance of orographic factors in the distribution of lichens was assessed using principal component analysis (PCA) and multiple regression analysis. It was found that the absolute altitude and distance from the glacier are of the greatest importance for crustose lichens, while for fruticose lichens, the most critical factors were the slope exposure and steepness. Along the altitudinal gradient, the number of species decreased (from 88 to 25). The highest number of species (90) was recorded at distances of 0.1 to 1.0 km from the glacier edge, which is explained by the unstable species composition of areas recently released from under the glacier. The number of species in all groups generally decreased (from 81 to 52) with increasing slope steepness. With an increasing heat supply of slopes (on a gradient from northern to southern), the number of species steadily increased in all groups (from 39 to 75). The low sum of the explained variance values for the first two PCA components (21%) characterizes the specificity of the natural environment of polar deserts, where there is no leading environmental factor.

## 1. Introduction

To date, microclimatic factors have been the most extensively studied among all the factors influencing lichen distribution. These factors are largely determined by orographic parameters, in particular altitude, slope exposure and steepness, and are related to indicators of species distribution, such as their number and abundance (cover) [1,2]. The influence of slope steepness and exposure on the species composition and cover of lichens is discussed in several works [3,4,5]. It is noted that strong winds on windward slopes lead to the rapid drying of substrates, which is extremely unfavorable for lichen growth. Additionally, the exposure and steepness of slopes affects the ability of lichens to attach to the substrate. Daniëls [3] showed that, on windward or steep slopes, the wind carries away *Cladonia* and *Cetraria* thalli before they can attach. Elevation above sea level is a major factor in determining the mean annual and summer temperatures. This factor, together with the slope steepness, largely influences the distribution of saxicolous lichens [4]. Additionally, environmental variables, such as the relative humidity and precipitation, on which lichen species diversity and the diversity of growth forms depend, are also related to the absolute altitude [6,7,8]. It is assumed that, with increasing altitude, the ratio of different growth forms of these organisms is related to an increase in the proportion of crustose forms [9].

The distribution of lichens in the deglaciation zone is determined by the peculiarities of the development of ice-free areas by these organisms. According to the data reported by D. Fahzelt et al. [1], the first settlers (saxicolous lichens) usually occupy the surfaces of boulders facing the opposite side of the glacier edge, most often occurring in cracks and caverns, protected from strong winds and temperature differences. Cracks located on the surface of large boulders are noted as being the most preferable for lichens. The slope angle and surface exposure largely determine the microclimatic regime of certain substrates, particularly stony (rubble) and rocky substrates, through their warming, frequency of surface freeze–thaw cycles and water availability. Microclimate factors have a direct effect on the surface texture of substrates, i.e., their smoothness, hardness and relative stability [10]. It is believed that lichen diaspores can become trapped and develop on rough surfaces more easily than on smooth ones. As a result, the species diversity of lichens on rough surfaces is higher.

Lichens often colonize areas freed from glacial cover as part of biological soil crusts together with cyanobacteria, algae, microfungi and bryophytes [11]. Established dependences of the species composition of vascular plants on the distance to the edge of the modern glacier and the degree of soil crust [12] development suggest that the composition of lichens also does not remain constant at different distances from the glacier. The territory adjacent to the glacier is believed to display a certain sequence (“zonality”) of vascular vegetation and lichens in relation to the retreating ice mass [1,13]. The growth rate (and finally the cover) of lichens is positively influenced by the proximity of the seashore colony of birds. Thus, according to R. L. Smith [14], the colonization and development of communities in the coastal lowlands of marine Antarctica, near sea level and in habitats influenced by seabirds, are relatively rapid processes.

The existing literature data on the influence of relief factors on the distribution of lichens in the Arctic is apparently not complete. In several works, the slope steepness and exposure, altitude and distance from the glacier are considered as independent variables. In our work, we are trying to analyze the joint influence on the number of species and the cover of lichens of several relief factors, namely, the altitude, distance to the glacier, slope steepness and exposure. For this purpose, we took the following steps: (1) to identify the most informative environmental factors for lichens of different growth forms using PCA methods and multiple regression analysis; (2) to study changes in the number of species and cover of lichens in different ranges and gradations of four orographic factors; (3) using paired regression models, to show how the cover of some lichen groups changes when the parameters of orographic factors change; (4) to build biplot diagrams and to use them for the identification of conjugate groups of variable types; (5) to give an interpretation of the observed dependencies.

### The Natural Conditions of the Research Area

The Franz Josef Land archipelago is one of the northernmost territories in the world. Administratively, it is a part of the Arkhangelsk Region of Russia and located in the Arctic Ocean between the archipelagos of Spitsbergen, Novaya Zemlya and Severnaya Zemlya (79–82° N, 44–68° E) (Figure 1). It includes more than 190 islands with a total area of 16,134 km^2^. The climate in the archipelago is typically Arctic. Despite the general warming trend, the weather in the archipelago is relatively stable, with only a slight softening of the short summer conditions [15]. The cold period lasts 8–10 months, and the average annual temperature does not exceed −12 °C, but in July and August, the average monthly temperature rises to +1.5–2.0 °C. The total duration of stable positive temperatures varies from 60–65 days in the south of the archipelago to 40–45 days in the north. The sun does not set throughout the summer on Franz Josef Land; however, more than 80% of the time, it is covered by clouds. The amounts of solar radiation are 25–26 kcal in June, 25 kcal in July and 15–16 kcal in August. The air saturation with moisture is extremely high and reaches 94% in August. The amount of annual precipitation varies on different islands from 195 mm to 300 mm, with approximately 85% of this amount being solid precipitation. Eastward and southeastward transport with a significant southerly component prevails on an annual average [15,16,17].

About 85% of the territory is currently under glaciers [16]. Large ice domes and sheets develop most intensively in the eastern part of each island [17]. Glaciers on many islands leave free only a small strip of land, often no more than 0.5–0.7 km wide, which is the main arena of life [18]. Many glaciers in the archipelago are now actively retreating. According to Milovanova et al. [19], over the last 50 years, the area of land unoccupied by glaciers in the archipelago has increased by 9%. Surface moisture in ice-free areas is provided mainly by glacial meltwater. The general pattern of moisture in the territory is largely determined by the close occurrence of permafrost. Moisture is retained in the depressions of the coastal plain, often under mosses. The active role of glaciers, snow and melt water in the formation of the modern landscape of the archipelago leads to the diversity of landforms, which is manifested at different levels, i.e., meso-, micro- and nanorelief.

The average altitude of the highest islands reaches 400–490 m (hereinafter, the altitude above sea level), and the maximum altitude is 620 m (Hooker Island). However, most of the rises of these islands (as a rule, plateau-shaped peaks) are covered with ice domes, the thickness of which is 100–200 m [17]. Retreating glaciers currently leave a hilly ridge topography with an altitude not exceeding 70–80 m (Figure 2). The degree of relief development also depends on the time of glacier melting on a particular island. In the case of relatively old glacier retreat, modern relief-forming processes manifest on a rather large territory (if the island is large enough). For example, on Alexandra Land, the edge of modern glaciers is 10–12 km away from the most remote land areas. All forms of meso- and microrelief found on the archipelago can be traced precisely in such nonglacial spaces.

In most land areas where the glacier has moved away from the seashore by 0.8–1.2 km, a seaside accumulative plain is formed (flat or hilly ridge with lakes) composed of either loamy-rubble material or sands. In some cases, when the glacier edge is located several hundreds of meters from the coast, several mesorelief elements are expressed in the relief of the land free from the glacier cover: a marine abrasion-accumulative terrace raised to an altitude of 20–25 m; a slope of the basalt plateau with remnants in the upper part (sometimes with cliffs–steep ledge walls with rookery) and a rather powerful cloak of coarse clastic rubble in the lower part; moraines (most often, headwaters), usually “carried” to the seashore and representing a chaotic pile of coarse clastic material; saddles between neighboring sections of the basalt plateau. The small islands in the straits are plateaus with an upper surface not occupied by ice cover and located at an altitude of 30–50 m.

The primary processes of mesorelief formation are superimposed by secondary ones, leading to the formation of numerous forms of micro- and nanoreliefs. The main forms of micro- and nanorelief that lichens colonize are given in Table 1.

## 2. Results

A total of 111 species of lichens were identified; of them, 47 species were crustose lichens, 35 were foliose lichens and 29 were fruticose lichens. All three lichen groups showed a decrease in the number of species (from 88 to 25 for all) with increasing altitude (ranging from 1 to >60 m), with the sharpest gradient in the case of crustose forms: from 34 taxa in the range of 1–20 m to 7 taxa at altitudes above 60 m (Table 2). The distribution of the total cover for crustose lichens was characterized by a unimodal curve with a maximum (21.7%) in the range of 41–60 m; for foliose lichens, by a decrease (from 45.5% in the range of 1–20 m to 18.3% in the range > 60 m); for fruticose lichens, by a unimodal curve with a maximum (67.9%) in the range of >60 m (Figure 3a). The highest average values of cover in the range of 1–20 m were found in three species, two of which belong to the group of foliose lichens (*Umbilicaria arctica* (Ach.) Nyl., *U. cylindrica* (L.) Delise) and one to fruticose lichens (*Pseudephebe minuscula* (Nyl. ex Arnold) Brodo & D. Hawksw.) (15.3%, 15.0% and 17.7%, respectively). The highest values of cover in the range of 21–40 m were found in two fruticose lichens, *Stereocaulon alpinum* Laurer and *S. botryosum* Ach.: 5.3% and 4.8%. In the range of 41–60 m, the cover of the latter species remained relatively high (5.0%), which was also observed in the foliose lichen *Flavocetraria cucullata* (Bellardi) Kärnefelt & A. Thell. This species had the highest cover (5.3%) at altitudes >60 m; in addition, the cover of the fruticose *Bryocaulon divergens* (Ach.) Kärnefelt (4.1%) and the crustose lichen *Ochrolechia frigida* (Sw.) Lynge (3.5%) increased in the range of >60 m (Table 2).

The general trend of a decrease in the number of species (from 90 to 41 for all) with increasing distance from the edge of the glacier can be traced for all three groups of lichens, although it breaks down in the range of 2.1–4.0 km, where the number of species increases. The largest number of crustose lichen species was noted near the edge of the glacier (34 species), while only 17 species were found at the maximum distance from the glacier (Table 2). The total cover of all groups of lichens was characterized by a wave-like distribution with two or three peaks: for crustose lichens, the maximum (26.2%) corresponded to the range of 8.1–12.0 km; for foliose lichens (45.6%), to the range of 0.1–1.0 km; for fruticose lichens (45.5%), to the range of 2.1–4.0 km (with similar values in the ranges 4.1–8.0 and 8.1–12.0 km) (Figure 3b). High values of cover in the immediate vicinity of the glacier (0.1–1.0 km) were found in *Umbilicaria arctica*, *U. cylindrica* and *U. torrefacta* (Lightf.) Schrad. (30.0%, 15.0% and 8.0%, respectively), as well as in *Pseudephebe minuscula* (25.0%). Only one species, *Stereocaulon botryosum*, had a relatively high cover (4.6%) in the range of 1.1–2.0%. In the range of 2.1–4.0 km, there was a relatively high cover of two foliose species, namely, *Cetrariella delisei* (Bory ex Schaer.) Kärnefelt & A. Thell and *Umbilicaria hyperborea* (Ach.) Hoffm. (8.8% and 5.9%, respectively), as well as one fruticose, *Stereocaulon botryosum* (8.0%). In the range of 4.1–8.0 km, the leaders were two fruticose species: *Cladonia stricta* (Nyl.) Nyl. and *Stereocaulon alpinum* (6.0% and 7.5%, respectively), and one foliose (*Cetrariella delisei*, at 5.5%). In the range of 8.1–12.0%, a cover of 4.1% was noted in the crustose species *Baeomyces carneus* Flörke; *Ochrolechia frigida* (4.4%) also belongs to the same group; however, the highest value of cover at the greatest distance from the glacier was achieved by the fruticose species *Stereocaulon alpinum* (11.5%) (Table 2).

The number of species generally decreased (from 81 to 52 for all) as the steepness of the slope increased, and this trend was most clear for the crustose lichens. The number of species reached 30 on horizontal and subhorizontal surfaces, and with a slope steepness of more than 20°, it decreased to 18 (Table 2). With a wave-like change in the total cover of crustose lichens, this value reached a maximum (25.1%) in the range of 11–20°. Two other groups—foliose and fruticose lichens—were characterized by a close to unimodal distribution of this indicator. Foliose lichens showed highest cover (52.5%) in the range of 1–5°, and fruticose in the range > 20° (59.4%) (Figure 3c). In the horizontal plots, there were two foliose species (*Cetrariella delisei* and *Umbilicaria cylindrica*) with a relatively high average cover (6.5% and 7.8%, respectively), and two fruticose species, *Sphaerophorus fragilis* (L.) Pers. and *Stereocaulon botryosum* (with a cover of 5.7% and 6.0%, respectively). Only one species, *Cetrariella delisei*, had an average cover of 5.0% in the slope steepness range of 1–5°. No one species had a relatively high cover in the slope steepness range of 6–10°. *Stereocaulon alpinum* was characterized by a high cover (11.0%) in the range of 11–20°. In the range of ˃20°, two species had high cover: foliose *Umbilicaria torrefacta* (8.0%) and fruticose *Pseudephebe minuscula* (12.8%) (Table 2).

The tendency for an increase in the number of species with an increase (from 39 to 75 for all) in the heat supply of the slope was manifested in all three groups of lichens. It was most pronounced for crustose forms: where, on the slopes of the northern and northeastern exposure, the number of species was 12, and on the southern and southwestern ones, it was 31 (Table 2). Crustose lichens showed a unimodal distribution of total cover, with a peak on the slopes of western–northwestern exposure (24.2%). The other two groups, foliose and fruticose, were characterized by a nonlinear change in cover: the cover of the first increased along the gradient from cold to warm exposure (from 14.3% to 53.1%), and of the second, it decreased in the same direction (from 70.9% to 35.6%) (Figure 3d). The highest values of cover on the slopes of northern–northeastern exposure included the species of foliose and fruticose growth forms (*Cetrariella delisei*—6.5%, *Flavocetraria cucullata*—8.8%, *Cladonia pyxidata* (L.) Hoffm.—4.3%). Foliose *Umbilicaria arctica*, with 12.0% cover, was prominent on the slopes of southern exposure. An intermediate exposure (east–southeast) was characterized by two foliose species, with a significant excess of cover in relation to the species of the coldest and warmest exposure—*Umbilicaria torrefacta* (8%) and *Pseudephebe minuscula* (12.0%) (Table 2).

There was only one crustose species, *Ochrolechia frigida*, with a tendency to change cover according to several features—absolute altitude, distance to the glacier and slope exposure. In the first case, there was an increase in cover up to altitude above 60 m; in the second case, the same increase in cover at the maximum distance from the glacier (8.1–12.0 km). In terms of the slope exposure for this species, there was a slight increase in cover on the slopes of neutral exposure (west, east) and the same slight decrease in cover on south and southwest slopes.

There were at least three foliose lichens that demonstrated distinct changes in cover according to three traits. In *Cetrariella delisei*, this parameter decreased with altitude, with increasing distance from the glacier edge (starting at a distance of 2.1 km) and with increasing slope steepness. The species *Flavocetraria cucullata* was characterized by a decrease in cover starting at a distance of 2.1 km from the glacier, and a unimodal dependence (with a sag of the regression line in the middle part) of cover on the altitude and slope exposure. *Foveolaria nivalis* (L.) S. Chesnokov et al. was characterized by decrease in cover with altitude, and the unimodal distribution (with a peak in the middle part) of this parameter according to slope steepness and exposure.

Only one fruticose species, *Stereocaulon botryosum*, showed significant differences in the distribution of the three parameters: its cover increased to a maximum in the altitude range of 41–60 m, decreased with an increasing slope steepness up to 10°, and had a unimodal dependence (with a sag in the middle) on the slope exposure.

Similar dependencies, but only for one or two parameters, were noted for some other species: *Cetraria islandica* (L.) Ach. showed a decrease in cover as the slope steepness increased, *Bryocaulon divergens* and *Thamnolia vermicularis* (Sw.) Schaer. showed an increase in cover with altitude and *Stereocaulon alpinum* showed an increase in this index as the distance from the glacier increased. The decrease in cover by altitude was characteristic for the species *Cetrariella delisei*, *Foveolaria nivalis* and *Umbilicaria torrefacta*, and an increase (starting from the range 21–40 m) was characteristic for *Flavocetraria cucullata*. Cover decreased with distance from the glacier (starting from the distance range of 2.1–4.0 km) for *Cetrariella delisei* and *Flavocetraria cucullata*, and for the first of these species, also with increasing slope steepness (when the slope steepness changed from the horizontal surface to a value of ˃20°).

For several species, high values of cover in only one range of a factor were revealed, with no or very insignificant (+) cover of these species in other ranges. Thus, several species of the genus *Umbilicaria* (*U. arctica*, *U. cylindrica*) had high values of cover in the first ranges of absolute altitude (15.3% and 15.0%) and distance to the glacier (30.0% and 15.0%, respectively); *U. arctica* was characterized by high cover (12.0%) on slopes of southwestern exposure; *Pseudephebe minuscula* had the same cover on slopes of eastern and southeastern exposure. *Cetraria muricata* (Ach.) Roum. had 5.0% cover on slopes of north–northeast exposure.

The relationship shown in Figure 4a demonstrates that, on the marine gently steeply sloping plain, an increase in altitude up to 50 m was accompanied by a decrease in the cover of crustose lichens (*Lepraria gelida* Tønsberg & Zhurb., *L. neglecta* (Nyl.) Erichsen, *Lecanora epibryon* (Ach.) Ach., *Ochrolechia frigida* and *Rinodina turfacea* (Wahlenb.) Körb.) from 5% to 1% (a logarithmic trend). The cover for lichens of the same growth form (*Baeomyces carneus*, *Ochrolechia frigida*, *Parvoplaca tiroliensis* (Zahlbr.) Arup et al. and *Psoroma hypnorum* (Vahl) Gray) had an increased polynomial dependence to 6–8% with an increasing distance to the edge of the glacier at a distance of approximately 6 km, and with a further increase in this distance (up to 9 km), it again decreased to 3–4% (polynomial dependence) on the same form of mesorelief (Figure 4b). There was an increase in the polynomial dependence in the cover of fruticose lichens (*Alectoria nigricans* (Ach.) Nyl., *Bryocaulon divergens*, *Pseudephebe minuscula*, *Sphaerophorus fragilis* and *Stereocaulon botryosum*) up to 70% with an increase in the slope steepness to 30° on the low coastal plain (Figure 4c). A sharp decrease in the cover of fruticose lichens (*Alectoria ochroleuca* (Schrank) Nyl., *Cladonia pyxidata*, *Pseudephebe pubescens* (L.) M. Choisy, *Sphaerophorus fragilis* and *Stereocaulon alpinum*) was noted in areas under the nesting grounds of sea colonial birds when the slope exposure changed from western to eastern through southeast to south: from 8–9% to 1% (a linear trend) (Figure 4d).

The PCA applied for 10 factors resulted in eigenvalues of component vectors and the percent of explained dispersion for each component (Table 3). All factors had eigenvalues >1.0, which allowed all 10 factors to be taken into account in the analysis, in accordance with one of the criteria for selecting the number of factors subject to meaningful interpretation [20]. Noteworthy is the extremely small total explained variance of the first two factors: approximately 21% for crustose and fruticose lichens. Multiple regression analysis revealed that factors three and eight were the most informative for crustose lichens with relatively high values of the b* coefficient for the altitude (b* = 0.5691) and distance to the glacier (b* = 0.4655), respectively, while factors seven and two were the most informative for fruticose lichens with high values of this coefficient for the exposure (b* = 0.4974) and slope steepness (b* = 0.4131) indices (Table 4). For foliose lichens, only one component (out of ten) had a variable with a relatively high B coefficient value (b* = −0.4322), namely, the slope steepness.

On a biplot showing the distribution of species and relevés in the component space for crustose lichens (Figure 5a), several variables are clustered near axis 1 (negative values of factor loadings): *Rinodina mniaroea* (Ach.) Körb, *Rhizocarpon inarense* (Vain.) Vain., *Lepraria neglecta* and *L. gelida*, and near axis 2 (positive values), *Rhizocarpon cinereovirens* (Müll. Arg.) Vain. and *R. copelandii* (Körb.) Th. Fr., and the vector of *Ochrolechia frigida* is slightly offset from it. The *Caloplaca stillicidiorum* (Vahl) Lynge vector, weakly correlated with axis 1 (positive values), is of lesser importance. The diagram for fruticose lichens (Figure 5b) highlights at least five groups or distinct vectors. Three vectors, *Alectoria nigricans*, *Cladonia phyllophora* Hoffm. and *Pseudephebe minuscula*, form a group adjacent to axis 2 (positive values); the vector of *Cetraria aculeata* (Schreb.) Fr. and, to a lesser extent, *Sphaerophorus fragilis*, correlate with the negative values of this axis. Vectors of two species, *Cladonia pyxidata* and *C. pleurota* (Flörke) Schaer., are close to axis 1 (negative values); the second of them has low values of factor loadings. The vector of *Usnea sphacelata* R. Br. correlates with axis 1 (positive values) and with the low values of factor loadings—*Stereocaulon condensatum* Hoffm. Two vectors weakly associated with the axes stand out in the diagram: *Cetraria muricata* (close to axis 1: negative values) and the vector *Cladonia amaurocraea* (Flörke) Schaer. (located closer to axis 2: negative values). Equidistant from both axes is the vector of two species, *Thamnolia vermicularis* and *Pseudephebe pubescens*, located in the upper-left quadrant of the diagram.

## 3. Discussion

The decrease in the number of species with altitude is a common feature across all lichen groups, particularly pronounced in the last part of the gradient, i.e., altitudes above 60 m above sea level. This decline is attributed by the rather harsh microclimatic regime of the habitats at these altitudes, due to the proximity to the glacier and the increased wind speed. The altitudinal distribution of cover values of different groups of lichens is apparently caused by different reasons. The total cover of foliose lichens decreases with increasing altitude, which is expected due to the erosive impact of wind at higher altitudes, leading to damage to the largest thalli [21]. On the contrary, the cover of fruticose lichens at an altitude above 60 m increases. This is explained by the prevalence at these altitudes of flagstones with crevices, which are inhabited by lichen patches, most often comprising several species. These flagstones fill cracks between polygons (including stone polygons), as well as contact zones between slope strips with different granulometric compositions. The decline in the cover of crustose lichens with increasing altitude (Figure 4a) is explained by recent rock exposure from beneath the glacial cover at altitudes of 60–80 m, and corresponds with the initial stage of the process of colonization and the outgrowth of patches and cushions of lichens. Vectors that correlate with axis 1 in Figure 5a characterize the variation in cover with altitude for *Lepraria gelida*, *L. neglecta* and *Rhizocarpon inarense*. According to Kukwa and Zhurbenko [22], *Lepraria gelida* dominates lichen communities in the coastal strip of the Severnaya Zemlya archipelago. Apparently, the gravitation of this species to low altitudes is related to its preference for moist conditions of the coastal strip. At the same time, stable rubble–pebble polygons are formed in this zone, with a surface where lichen cover reaches 80% (Figure 6). *Rhizocarpon inarense*, indicative of these conditions, can grow under snow [23] and accumulates in deep (0.6–0.7 m) and narrow hollows and cracks between polygons.

The high number of species near the glacier edge in all three lichen groups is related to the availability of free and uninhabited relief elements and planes in the periglacial region. This is primarily characteristic of crustose lichens, which utilize the numerous cracks and caverns on clumps and boulder surfaces. On the other hand, the reason for the high number of species in the glacial region may result from the random selection of species at each site, contributing to a high total number of species. As one moves away from the glacier edge, lichen cover increases to 75%. Indicators of conditions at such sites are *Rhizocarpon cinereovirens* and *R. copelandii* (vectors correlated with axis 2 in Figure 5a). According to Węgrzin [24] and Malíček et al. [25], *Rhizocarpon cinereovirens* settles on rocks exposed to winds and free of snow for an extended period. We observed an increase in the wind force at a distance of 5–7 km from the glacier on several islands, such as George Land, Alexandra Land and others.

The highest number of species in all groups, observed on slopes with a steepness up to 5°, is due to the fact that it is in these areas that soils and polygons (fine-grained, fine gravel) most conducive to lichen establishment are formed. However, different ranges of slope steepness, in which the optimum (maximum cover) of one or another lichen group is manifested, indicate different reasons for the proliferation of lichens on these slopes. The relatively large cover of crustose lichens on steep slopes of 11–20° is due to the fact that species of this group prefer sloping slopes facing the direction opposite to the prevailing winds, where the erosive effect of winter winds is less pronounced. Foliose lichens predominate in subhorizontal areas, which are represented by numerous polygons (rubbly and loamy, polygon cells). Here, they settle on the leeward side with weak snow accumulation. The predominance of fruticose lichens on steep slopes, just as in the case of crustose lichens, is due to the need for protection from strong winds. The vectors of *Alectoria nigricans* and *Pseudephebe minuscula* in the biplot diagram (Figure 5b) are associated with a sample area laid out on a low marine terrace. The variation in cover from 6% (the surface of the polygons) to 65% (the cracks between the polygons) is determined by differences in the wind regime on these two relief elements, i.e., strong snowstorm transport with snow drift in the first case, and zones of wind calmness and snow accumulation in the second one.

All lichen groups are characterized by increases in the number of species as the slope exposure changes from northern to southern, i.e., as the heat availability of the slopes increases. This trend is most pronounced for crustose lichens, where the number of species reaches 31 on the slopes of southern and southwestern exposure. This dependence is evident not only on slopes but also on the slopes of freestanding large boulders, the southern side of which can be covered with crustose lichens by 65–70%. However, there is no such trend for the index of cover for all lichen groups. Total cover increases with increasing heat availability only in foliose lichens. Fruticose lichens show the opposite tendency—the highest values of their cover are observed on the coldest (northern) slopes, and cover decreases with increasing heat availability. Apparently, fruticose lichens thereby avoid higher heat input and consequently more rapid moisture evaporation on south-exposed slopes [26]. For the distribution of fruticose lichens and foliose lichens on slopes of different exposure, the nivality factor is of significant importance; for example, *Cetrariella delisei* patches grow intensively under a good snow shelter (assuming a long period of snowfall and the upper soil horizon in low temperatures). In this case, the shielding effect of flagstone is crucial, which detains strong southerly winds and promotes snow accumulation. The vector of *Cladonia pleurota* and *C. pyxidata*, which is correlated with horizontal axis 1 (Figure 5b), is associated with variation in the cover of these species in sample plots with high cover (up to 95%) of mosses. The latter often form tubercles 5–7 cm high above the general surface of mosses, and lichens settle on their dying areas. The thalli of the *Cladonia* species usually occupy different slopes from the knolls, including windward slopes, due to their resistance to wind and abrasion by ice particles [27]. The above two *Cladonia* species grow intensively in anthropogenically disturbed habitats. Thus, on Alger Island, at the site of the late 19th century expedition camp, on the south-facing side of logs, the total cover of these lichens was 5–6%. The sharp decrease in cover on slopes of southern exposure compared to neutral slopes (western, eastern and southeastern) is explained by the intensive growth of mosses, which displace lichens.

## 4. Materials and Methods

### 4.1. Lichen Collection and Laboratory Studies

The lichens were collected by the first author during a geobotanical survey of the territory in 2012. In total, about 500 lichen specimens were collected on 23 islands of the archipelago: Alger, Bryce, Wilczek, Wiener-Neustadt, Gall, Gage, Hoffmann, Greeley, Hooker, Jackson, Eva-Liv, Alexandra Land, Wilczek Land, Georg Land, Kane, Kuhn, La-Ronsjer, McClintock, Mejbel, Nansen, Northbrook, Ziegler and Champ (Figure 7). 

Identification of the lichen collection was carried out by the second and third authors using morphological features and standard color reactions (spot test) detected in 10% potassium hydroxide (KOH or K), sodium hypochlorite (C), K followed by C on the same fragment (KC) and paraphenylenediamine (PD) [28,29]. The nomenclature of species is given in accordance with the *Checklist of Fennoscandian Lichen-Forming and Lichenicolous Fungi* [30] and special papers [31,32]. The specimens were stored in the herbarium of the Komarov Botanical Institute of the Russian Academy of Sciences (LE). Figure 7 was prepared in the GIS program Axiom 5.1.

### 4.2. Geobotanical Research

Sample plots (4 m × 4 m) were laid out on land areas between the seashore and glaciers. Within one type of mesorelief, the most characteristic areas were selected for the test plots, where orographic factors corresponded to the entire type of mesorelief. For example, the sample plot on a marine terrace should be located in the part where the slope corresponds to the general slope and steepness of the whole terrace. In addition, the site should be as homogeneous as possible with respect to the moisture, snow accumulation, insolation and granulometric composition of soils. In some cases, a sample plot may be relatively heterogeneous if different microenvironmental elements, e.g., cracks between polygons or cells ca. 0.2 m × 0.2 m in diameter, are regularly repeated within its boundaries. In this case, the total lichen cover of the entire sample plot was considered, regardless of the slight variation in cover in different microhabitats within the same sample area (Figure 8). At each sample plot, we assessed the cover of lichens, mosses, liverworts and vascular plants. All relevés were made in the altitude range from sea level to an altitude of 88 m. The distance to the glacier was estimated based on satellite imagery. In most cases, the sample plots of the relevés in polar regions vary in the range of 3 m × 3 m–5 m × 5 m. The 4 m × 4 m sample plots we have chosen allowed us to compare the data obtained with data from other Arctic regions.

Altogether, 130 relevés were analyzed based on the main elements of the meso- and microrelief. Lichen cover was assessed visually in the field using estimates of 0.5%, 1.0%, 1.5%, 2.0%, 3.0%, etc., for each of the foliose and fruticose species. Small average cover values, e.g., 0.8 or 1.1, were obtained by averaging data from several sites. For example, if cover at several sites was rated consistently as 0.5, 0.5, 1.0, 1.0, then the average value was 0.8 (in Table 2, all values less than 1.0% are marked by ‘+’). When the cover was rated 1.0, 1.0. 1.5, 1.5, the average value was 1.3. This value was shown in Table 2 with the corresponding error of the mean. For crustose lichens, the total cover was assessed. If the total cover of crustose lichens was 1% or <1%, then all types of crustose lichens were noted by ‘+’. If the cover of crustose lichens was >1%, for example, 1.0%, 1.5%, 2.0%, then this value was divided proportionally between the identified species. The contribution of each species was determined by the ratio of species cover on individual stones or flagstones. In the field, samples of stones were selected on which there were quite a lot of crustose lichens, and the ratio of different species (identified by the color of the thalli) approximately corresponded to the ratio of species or groups of species on all other stones within a sample area of 4 m × 4 m. Thus, if the cover of crustose lichens on the entire sample area was estimated at 2%, and on individual stones 3 species were identified with corresponding cover of 50%, 30% and 20%, then the cover of these species was estimated as: 1%, ‘+’, ‘+’.

Easily identifiable lichens were not purposefully collected, but they are present in the collections as additional species. Crustose lichens and some species difficult to identify in the field were collected each in a separate envelope, where the site number, coordinates and the species cover were indicated.

We divided each of the orographic factors (altitude, distance to the nearest glacier, slope steepness and exposure) into several ranges. Four ranges (in meters) were adopted for the altitude parameter: 1–20, 21–40, 41–60, ˃60; for the distance to the glacier, five ranges were adopted (in km): 0.1–1.0, 1.1–2.0, 2.1–4.0, 4.1–8.0, 8.1–12.0; for the slope steepness, five gradations were adopted (in degrees): 0, 1–5, 6–10, 11–20, ˃20. The slope exposure factor was divided into 4 groups according to heat supply from the coldest to the warmest: N-NE, NW-W, E-SE, S-SW. The average value of the cover was calculated for each lichen species within the specified ranges. For example, the average cover was separately calculated for the altitude ranges of 21–40 m and 41–60 m, for the distance to the edge of the glacier of 2.1–4.0 km and 4.1–8.0 km, etc. The sets of sample plots for different ranges of both one factor and different factors varied; therefore, the average values of the cover of one lichen species in Table 2 are mutually independent. Slope steepness and exposure were determined for the habitat as a whole. The steepness of the slope was measured using a mountain compass, which was mounted on a long rod placed along the dip of the slope in the middle part of the sample plot. The orientation of the rod also served to determine the exposure of the slope. 

When assessing the cover on the sample plots with large boulders, we summed the cover of lichens growing on the main part of the plot, which is most often composed of crushed stone and loam, and the cover of lichens growing on the surface of boulders parallel to the general slope and the exposure of the 4 m × 4 m sample plot. For example, when assessing the cover on a site at the edge of a high sea terrace with a large number of boulders, we summarized the cover of species growing on the subhorizontal surface of the sample plot (in this case, 0–3°), the slope of which corresponded to the general slope of the coastal plain (in this case, to the south), and a cover of lichens growing on the flat tops of boulders of the same steepness and exposure. The cover of lichens located on other surfaces with a slope of 5–90° was not taken into account.

The idea of micro- and mesorelief forms as habitats, which are characterized by a certain set of plants and lichens, was accepted in this work. The division of relief forms according to their sizes into meso-, micro- and nanorelief forms was carried out in accordance with the Rychagov [33]. Each type of habitat corresponded to a certain set of direct environmental regimes, consisting of regimes of light and water supply, as well as heat and elements of mineral nutrition [34]. The set of such regimes was determined by the altitude above sea level, the steepness and exposure of the slope and several other factors. A certain group of lichens which represents a horizontal “cut” of the polar desert community, and which is part of a single moss–lichen layer, was isolated within each type of habitat.

### 4.3. Statistical Analysis

All lichens were allocated to three groups according to the growth form (crustose, foliaceous and fruticose) to analyze their distribution in the landscape. The PCA and multiple regression analysis were used in the data analysis. All calculations were carried out in the program Statistica 12. Standard error was calculated for all values of the mean.

The principal component analysis (PCA) method was used to identify the environmental factors that most strongly influenced the distribution of lichens in the archipelago landscape, as well as to assess the contribution of each of the 4 orographic factors to the overall variability in the number of species and the total cover of lichens. For 10 components (factors), both the values of factor loadings for variables (species) and the values of factors for objects (relevés) were obtained. The analysis was carried out differentially for each group of lichens. The ratio of the number of variables (p) to the number of objects (n) in lichen groups—47:113 (crustose), 35:117 (foliose) and 29:119 (fruticose)—corresponded to the recommendations for the ratio of these parameters (from 1:2 to 1:5) and the number of objects (50–80) for conducting factor analysis or principal component analysis in environmental studies [35].

When selecting the components that were most informative in terms of the 4 orographic factors considered in this paper, a multiple regression analysis model was used, and particularly important was the value of the standardized regression coefficient B. The independent variables (predictors) in the multiple regression equation were the measured values of altitude, distance to the glacier, slope steepness and exposure, and the dependent variables (response) were the factors (components)–vectors obtained by using the principal component method. This was based on the assumption that each component is interpreted by the same external variables [36]. The “step-by-step with inclusion” method was used. For each lichen group, a group of 4 predictors was tested relative to 10 factor axes (responses). Then, we analyzed the obtained values of the b* coefficient and selected 2 factors with the highest values of this index for the predictors occupying the first row among the predictors (after the row with statistics related to the free term) in the table of results of the multiple regression analysis. The statistical significance of the regression equation coefficients and the free term were assessed at a significance level of *p* = 0.1.

Two diagrams (biplots) were constructed for crustose and fruticose lichens, which reflected both groups of associated species and groups of relevés based on the results of using principal component analysis and multiple regression analysis. Relevés were represented by points and species by vectors, which were further interpreted in terms of their correlation with certain environmental factors in this case [37]. Scaling of the two data series was performed; in particular, the multiplication and division of factor loadings and factor values by some constants in order to better visualize the results of the principal component analysis. Factor (component) values for the objects (relevés) were standardized (the scales of both biplots varied between −1.0 and +1.0) [38]. We were guided by the following provisions in the biplot analysis: the length of the vector is equal to the variance of the corresponding variable; the angle between the vector and the axis indicates the importance of the contribution of the corresponding variable (species) to the principal component (component); the angle between pairs of vectors indicates the correlation between the corresponding variables (species); points located close in the diagram represent objects (relevés) with the same characteristics [39]. The horizontal and vertical axes were equalized with each other to make the ratio of the lengths of different vectors clearer. Vectors that almost coincided in length and direction (ending at points indistinguishable in the diagram) were represented by a single vector. The diagrams were the basis for analyzing groups of associated species, primarily those furthest from the zero point, and relevés located in roughly the same direction as the corresponding vectors. Vectors occupying an intermediate position between the axes, as well as vectors adjacent to the axes for which no acceptable solution was found, were left uninterpreted. The diagrams do not reflect points near the zero point.

We used the idea of a regression curve (diagrams are not given in the paper) formed by average values in the sequence of ranges corresponding to the increase in their absolute values when analyzing the distribution of average values of cover in different ranges and gradations (except for the slope exposure indicator, where the increase in the heat supply factor was significant only for the extreme elements of the gradation series).

## 5. Conclusions

The predominant trend evident in the landscape of the Franz Josef Land archipelago is the active release of the land from glaciers. Lichen colonization of the land is highly uneven and depends on numerous environmental factors, which, in addition to those discussed in this paper, include the proximity of the site to the sea, the intensity of colluvial and dealluvial processes, solifluction, the primary weathering of rocks, differences in summer temperatures in different parts of the archipelago and others. The retreat of glacial domes on many islands has only recently commenced. In many cases, the width of the strip between the seashore and the edge of the modern glacier is measured in the first hundreds of meters, within which some areas are still covered with ice, the topsoil is saturated with moisture and there are very few stable ground areas (except for bedrock outcrops). This complexity makes it extremely challenging to establish any trend in the dynamics of lichen cover formation at these sites. The surveyed islands have very few large ice-free land areas. The most extensive land area free of the glacier is only on Alexandra Land Island, where the largest distance to the glacier is about 12 km. As one moves away from the glacier edge, the influence of the proximity of the sea coast begins, resulting in a gradient of distance to the glacier being superimposed on the gradient of the distance to the sea coast. Ecological gradients of a global order, particularly latitudinal zonality and altitudinal zonality, are not sufficiently distinct in the archipelago. The necessary range of altitudes is “lacking” for the manifestation of the altitudinal belt. On many islands, the glacier edge occurs above the 100 m sea level, leading to the interruption of the altitudinal belt series. All these factors contribute to the pronounced global ecological patterns on the archipelago.

The above analysis of lichen distribution along gradients of orographic factors reveals the absence of sufficiently strong relationships of cover indices with absolute altitude, the distance to the glacier edge, etc. It is often impossible to assess the contribution of several factors, such as the absolute altitude and the distance to the glacier edge, to the distribution of lichens on relief elements, as they are closely interrelated—the lower the site is located (down to the water edge), the farther it is from the glacier. Relatively high values of the R2 coefficient in pairwise regression relationships can be obtained only by strongly reducing the sample. Such dependencies are well revealed within area-limited mesorelief forms, such as a high abrasion terrace, a low marine terrace adjacent to a lagoon, etc. For large samples that include a significant portion of the islands of the archipelago, such as the southern or western islands, it is virtually impossible to identify a leading environmental factor responsible for the distribution of lichens throughout the area. This is confirmed by the low values of the eigenvalues of the first two factors (components) in the above principal component analysis: in two cases—for crustose lichens and fruticose lichens—their total share in the total variance is about 21% (Table 2). Similar values of the factor of the distance to the glacier (11% of the total dispersion for the first factor) were obtained for vascular plants of the polar deserts of Ellesmere Island in the Canadian Arctic [12]. It can be assumed that the absence of the leading environmental factor is generally characteristic of the entire polar desert zone.

## Figures and Tables

**Figure 1 plants-13-00193-f001:**
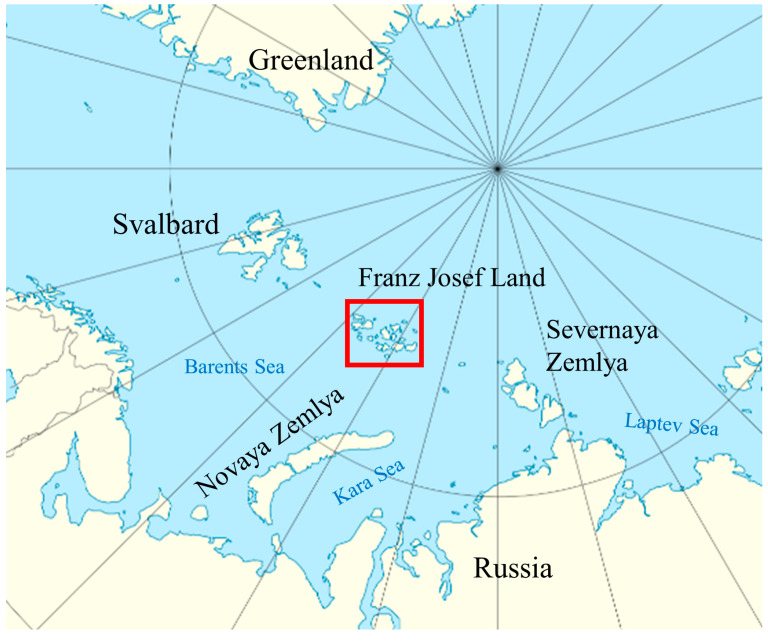
Location of the study area. Red square indicate the location of the study area; Franz Josef Land Archipelago.

**Figure 2 plants-13-00193-f002:**
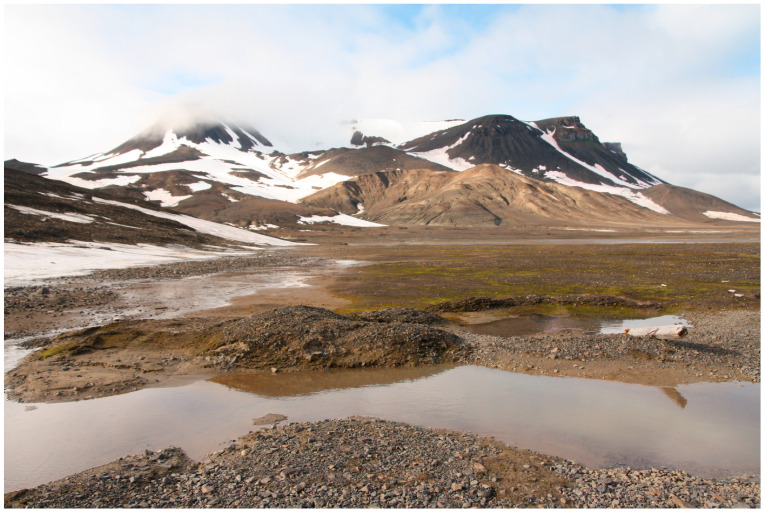
Hilly ridge topography after the disappearance of the glacier, Wiener Neustadt Island.

**Figure 3 plants-13-00193-f003:**
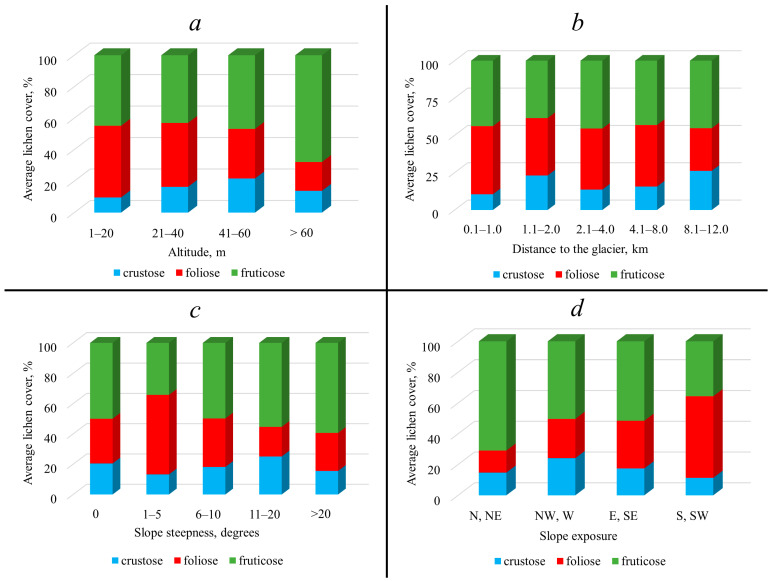
Average total cover of lichen life forms in different ranges (%): (**a**) altitude; (**b**) distance to the glacier; (**c**) slope steepness; (**d**) slope exposure.

**Figure 4 plants-13-00193-f004:**
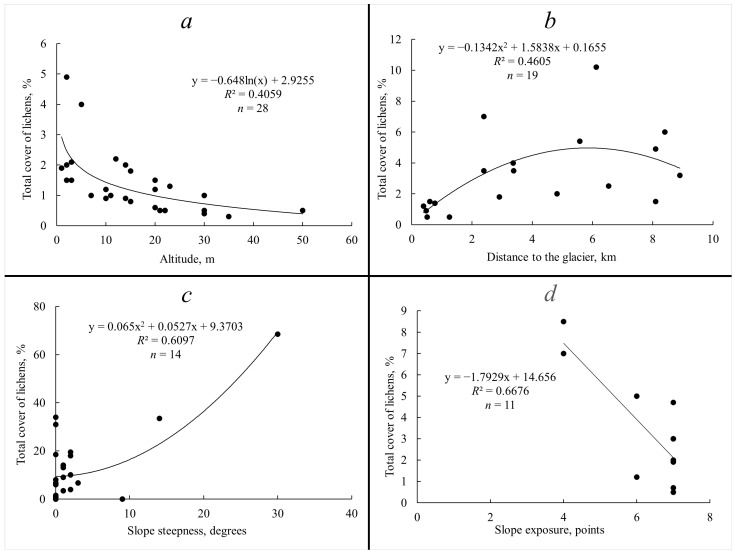
The relationship between the cover of: (**a**) crustose lichens and altitude; (**b**) crustose lichens and the distance to the glacier on a marine accumulative gently undulating plain; (**c**) fruticose lichens and slope steepness on a low coastal plain; (**d**) fruticose lichens and slope exposure in areas under the nesting grounds of sea colonial birds.

**Figure 5 plants-13-00193-f005:**
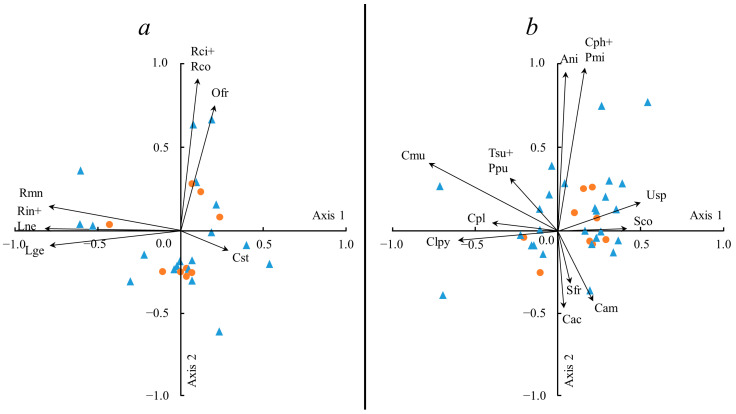
Species and sample plots in the space of the main components: (**a**) crustose lichens; (**b**) fruticose lichens. Vectors—species: Cst—*Caloplaca stillicidiorum*, Lge—*Lepraria gelida*, Lne—*L. neglecta*, Ofr—*Ochrolechia frigida*, Rci—*Rhizocarpon cinereovirens*, Rco—*R. copelandii*, Rin—*R. inarense*, Rmn—*Rinodina mniaroea* (**a**), Cam—*Cladonia amaurocraea*, Cph—*C. phyllophora*, Cpl—*C. pleurota*, Clpy—*C. pyxidata*, Cac—*Cetraria aculeata*, Cmu—*C. muricata*, Gni—*Alectoria nigricans*, Pmi—*Pseudephebe minuscula*, Ppu—*P. pubescens*, Sfr—*Sphaerophorus fragilis*, Sco—*Stereocaulon condensatum*, Tsu—*Thamnolia vermicularis*, Usp—*Usnea sphacelata* (**b**); circles—other species, triangles—sample plots of relevés.

**Figure 6 plants-13-00193-f006:**
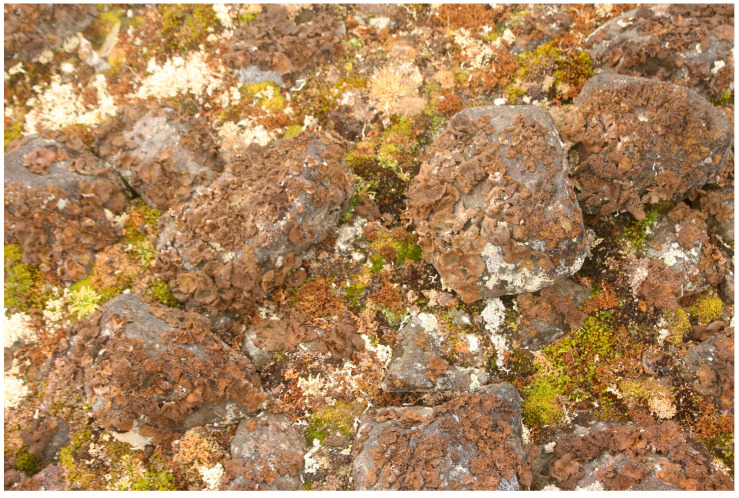
Sample plot with rubble–pebble polygons, where lichen cover reaches 80%.

**Figure 7 plants-13-00193-f007:**
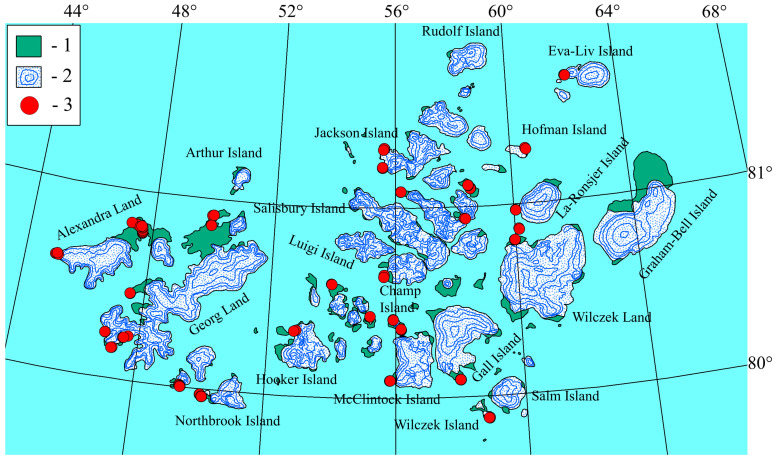
Location of sample plots of relevés on the Franz Josef Land archipelago. 1—ice-free areas; 2—glaciers; 3—sample plots of the relevés.

**Figure 8 plants-13-00193-f008:**
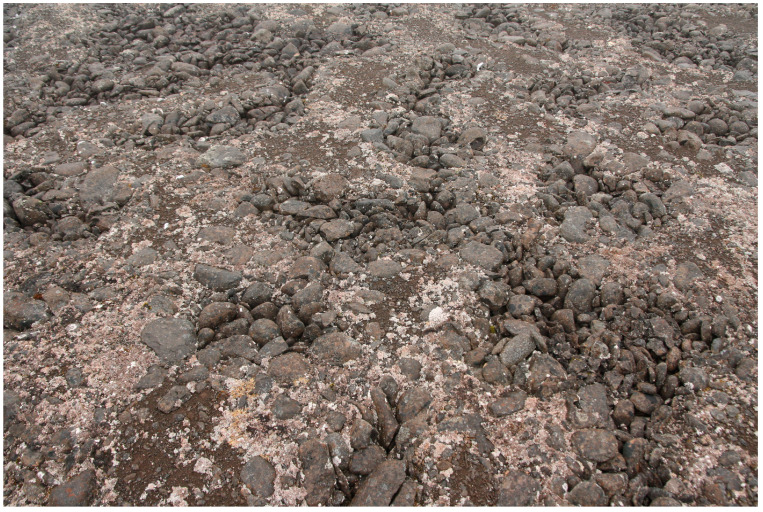
Sample plot with polygons, where the cover of *Stereocaulon* species on polygons and cracks between polygons is different.

**Table 1 plants-13-00193-t001:** The main forms of micro- and nanorelief of the islands of the Franz Josef Land archipelago.

Name\Parameter	The Shape of the Mesorelief, within Which the Microrelief Is Formed	The Steepness of the Slope Elements, Degrees	Diameter, m	The Excess of Relief Elements over Each Other, m	Granulometric Composition, Processes of Rock Destruction	Nanorelief Elements, Secondary Polygons, Diameter, m	Elements of Micro/Nanorelief, to Which Lichens Are Confined
Stony–gravelly polygons	High-sea and postglacial plains	0–3	0.6–1.2	0.1–0.3	Small rubble and gravel	0.2–0.3	Contact zones of the polygon and stone rim
Polygons–meshes	Marine terraces of different levels	0	0.2–2.0	–	Small boulders	–	Small boulders in meshes and depressions
Fine-grained and gravelly polygons	Accumulative plains with flattened surfaces	0	0.2–0.3	–	Sandy loam, light loam; fine crushed stone, gravel	–	Cracks between polygons
Gravelly polygons	Low marine terraces	35–40	25–40	0.7–0.8	Rubble	5–15	Cracks between polygons
Loamy polygons	Marine or postglacial accumulative hollow–humped plains	1–3	0.15–0.35	0.02–0.05	Medium loam	–	Cracks between polygons
Spots in the moss carpets	Marine-accumulative (hilly ridge) and abrasive-accumulative plains	0–5	0.3–1.0	0.02–0.04	Medium or heavy loam	–	Contact areas of spot and moss cover
Spots in the liverwort carpets	Marine-accumulative (hilly ridge) and abrasive-accumulative plains	0–5	0.8–1.2, irregular shaped spots	0.03–0.06	Sand, sandy loam, light loam with an admixture of fine crushed stone	–	Contact areas of spot and liverwort carpet
Slope strips	Slopes of basalt plateaus	15–25	0.2–1.0	0–0.15	Loam (light and medium), crushed stone; flagstone, small blocks	–	Contact zones between strips with different granulometric composition
Slope steps	Slopes of basalt plateaus	30–40—the main surface; 10–12—terrace platform-steps	0.3–1.0	0.4–0.7	Medium loam, crushed stone, flagstone	–	The edge part of the steps, the side of the underlying step
Loam–peat mounds	Sea-accumulative plains	1–35	3.0–4.0	0.5 (exceeding the top over the foot)	Peat, medium loam; formation of frost-breaking cracks	Small peat fragments	Top, slopes and foot of a peat hillock
Blocks of basalt	Marine abrasive-accumulative terraces	0–90	1.0–2.5	1.2–1.5 (exceeding the top over the foot)	Basalts; flaking of rock fragments, laying of cracks	–	Cracks in the rock

**Table 2 plants-13-00193-t002:** Number of species and average values of the cover of lichens (%) on the sample plots in different ranges of orographic factors.

	Altitude, m				Distance to the Glacier, km					Steepness Range, Degrees					Slope Exposure			
	1–20	21–40	41–60	>60	0.1–1.0	1.1–2.0	2.1–4.0	4.1–8.0	8.1–12.0	0	1–5	6–10	11–20	>20	N, NE	W, NW	E, SE	S, SW
**Crustose lichens**																		
*Agonimia gelatinosa* (Ach.) M. Brand & Diederich		+					+						+					+
*Arthrorhaphis alpina* (Schaer.) R. Sant.		+			+	+				+	+							+
*Baeomyces carneus* Flörke		1.0 ± 0.7			+	+			4.1 ± 2.8	4.3 ± 3.2	+	+				+	1.0 ± 0.7	+
*Biatora ementiens* (Nyl.) Printzen	+	+	+		+		+	+		+	+					+		+
*Blastenia ammiospila* (Wahlenb.) Arup et al.	+	+			+		+			+	+	+				+	+	+
*Bryoplaca jungermanniae* (Vahl) Søchting et al.	+	+			+			+		+								
*Caloplaca stillicidiorum* (Vahl) Lynge	+	+	1.0 ± 0.3		+	+	+		+	+	+	+		1.0 ± 0.3			+	+
*Candelariella aurella* (Hoffm.) Zahlbr.	+	+	+		+			+		+				+			+	
*C.* cf. *canadensis* H. Magn.	1.1 ± 0.7					+	2.0 ± 0.6			2.0 ± 1.3	+							1.1 ± 0.7
*Helocarpon crassipes* Th. Fr.	+	+			+		+			+	+		+		+			+
*Japewia tornoënsis* (Nyl.) Tønsberg	+						+				+							+
*Lecanora epibryon* (Ach.) Ach.	+	+	1.0 ± 0.4		+		+	2.0 ± 0.7	+	+	+	+		1.0 ± 0.7		+	1.0 ± 0.6	+
*L. polytropa* (Ehrh. ex Hoffm.) Rabenh.	+	+	+		+		+		+	+				+			+	
*Lecidea lapicida* (Ach.) Ach.	+		+		+							+		+			+	+
*L. ramulosa* Th. Fr.	+	+				+		+		+	+					+		
*Lepraria caesioalba* (B. de Lesd.) J. R. Laundon	+						+			+								
*L. gelida* Tønsberg & Zhurb.	+	+			1.1 ± 0.5	+	+	+	+	+	+	+		1.0 ± 0.7	+	+	+	+
*L. neglecta* (Nyl.) Lettau	+			+	+	+	+			+	+	+		+			+	+
*Megaspora verrucosa* (Ach.) Hafellner & V. Wirth		+			+		+		+				+			+	+	+
*Micarea incrassata* Hedl.	+	+							+	+	+		+			+		+
*M. lignaria* (Ach.) Hedl.		+			+		+						+				+	+
*Miriquidica lulensis* (Hellb.) Hertel & Rambold			+		+									+			+	
*Ochrolechia frigida* (Sw.) Lynge	2.2 ± 0.4	2.5 ± 0.6	2.6 ± 1.8	3.5 ± 2.3	1.2 ± 0.2	1.7 ± 0.9	3.2 ± 1.4	3.8 ± 0.7	4.4 ± 1.3	3.2 ± 0.7	2.0 ± 0.4	1.7 ± 0.5	3.9 ± 2.8	1.0 ± 0.2	1.9 ± 1.0	2.9 ± 1.5	2.1 ± 0.6	1.6 ± 0.4
*Parvoplaca tiroliensis* (Zahlbr.) Arup et al.	+	+	1.0 ± 0.4	+	+		+	+	+	+	+	+	+	+	+	+	+	+
*Pertusaria geminipara* (Th. Fr.) C. Knight ex Brodo	+	2.0 ± 1.4				1.6 ± 1.3			1.0 ± 0.4	+	2.0 ± 1.4	1.0 ± 0.4				2.0 ± 1.4		1.0 ± 0.4
*Polyblastia gothica* Th. Fr.		+							+			+				+		
*Porpidia melinodes* (Körb.) Gowan & Ahti		+					+						+					+
*Protomicarea limosa* (Ach.) Hafellner	+				+		+			+	+					+		
*Protopannaria pezizoides* (Weber) P. M. Jørg. & S. Ekman	+	+	+	+	+	+	+	+		+	+	+	+	+	+	+	+	+
*Protothelenella sphinctrinoidella* (Nyl.) H. Mayrhofer & Poelt		+	+		+		+			+								
*Psoroma hypnorum* (Vahl) Gray	+	1.0 ± 0.4	+	+	+	+	+	+	1.3 ± 0.8	+	+	+	1.8 ± 1.2	+	+	1.4 ± 0.6	+	+
*Rhizocarpon cinereovirens* (Müll. Arg.) Vain.	1.0 ± 0.7						1.0 ± 0.5			1.0 ± 0.6								
*R. copelandii* (Körb.) Th. Fr.	+			+	+		+			+				+				+
*R. geminatum* Körb.	1.0 ± 0.6	+	+		+				+		+			+	+		+	
*R. geographicum* (L.) DC.			+		+									+			+	
*R. inarense* (Vain.) Vain.	1.0 ± 0.6	+			1.0 ± 0.6				+		+			1.0 ± 0.3	+		1.0 ± 0.5	
*Rhizoplaca melanophthalma* (DC.) Leuckert & Poelt		+	+		+	+					+			+		+	+	
*Rinodina mniaroea* (Ach.) Körb.	+						+				+							+
*R. olivaceobrunnea* C. W. Dodge & G. E. Baker	+	+	+		+	+	+			+	+	+	+			+	+	+
*R. terrestris* Tomin	+					+						+					+	
*R. turfacea* (Wahlenb.) Körb.	+	+		+	+	1.0 ± 0.6	+	+	+	+	+	+	+		1.4 ± 0.9	+	+	+
*Rostania ceranisca* (Nyl.) Otálora et al.	+	+			+	+			+	+	+						+	+
*Sporastatia testudinea* (Ach.) A. Massal.							+			+								
*Tetramelas insignis* (Nägeli ex Hepp) Kalb	+	+	+		+	1.0 ± 0.4	+			+	+		+		1.0 ± 0.3	+	+	+
*T. geophilus* (Flörke ex Sommerf.) Norman	+				+						+				+			+
*T. papillatus* (Sommerf.) Kalb			2.0 ± 0.9		2.0 ± 0.8						2.0 ± 0.9							2.0 ± 0.9
*Tremolecia atrata* (Ach.) Hertel	1.0 ± 0.7	+			1.0 ± 0.6		+		+		+		+	1.0 ± 0.4	+		1.0 ± 0.8	+
Number of crustose species	34	32	18	7	34	17	29	11	17	30	30	16	14	18	12	19	26	31
**Foliose lichens**																		
*Arctocetraria nigricascens* (Nyl.) Kärnefelt & A. Thell	1.0 ± 0.6				+	1.0 ± 0.4					1.0 ± 0.6			+	1.0 ± 0.6		+	
*Cetraria ericetorum* Opiz	+	1.0 ± 0.6				+	1.3 ± 0.8	1.0 ± 0.6	+	1.0 ± 0.6	+				+	+		1.3 ± 1.0
*C. islandica* (L.) Ach.	2.2 ± 0.3	2.3 ± 0.5	2.0 ± 1.2	1.9 ± 0.8	2.1 ± 0.4	1.1 ± 0.5	2.4 ± 0.5	3.2 ± 1.1	2.2 ± 0.9	2.6 ± 0.5	2.1 ± 0.3	1.7 ± 0.5	1.6 ± 0.6	+	3.3 ± 1.8	1.6 ± 0.6	1.8 ± 1.2	1.9 ± 0.3
*Cetrariella delisei* (Bory ex Schaer.) Kärnefelt & A. Thell	6.5 ± 1.6	3.8 ± 1.7	1.0 ± 0.6	+	3.0 ± 1.5	2.3 ± 1.1	8.8 ± 2.8	5.5 ± 2.1	3.1 ± 1.1	6.5 ± 2.0	5.0 ± 1.7	2.5 ± 1.3	1.4 ± 0.7	1.0 ± 0.3	6.5 ± 3.7	1.3 ± 0.6	2.7 ± 1.7	4.9 ± 2.2
*C. fastigiata* (Delise ex Nyl.) Kärnefelt & A. Thell							+			+								
*Flavocetraria cucullata* (Bellardi) Kärnefelt & A. Thell	3.7 ± 0.6	2.8 ± 0.4	5.0 ± 3.1	5.3 ± 1.1	3.7 ± 0.7	1.7 ± 1.0	3.8 ± 0.7	3.4 ± 1.2	2.1 ± 0.9	3.3 ± 0.6	3.8 ± 0.7	2.1 ± 0.6	4.5 ± 2.7	3.0 ± 0.7	8.8 ± 3.3	2.3 ± 1.1	2.1 ± 1.3	3.4 ± 0.6
*Foveolaria nivalis* (L.) S. Chesnokov et al.	3.5 ± 1.5	2.6 ± 1.2	1.3 ± 0.9		3.3 ± 1.5	3.0 ± 2.2	2.6 ± 1.6	4.2 ± 2.2	1.8 ± 1.2	3.0 ± 1.2	3.9 ± 2.4	2.3 ± 1.5	1.5 ± 0.7	1.3 ± 1.0	2.9 ± 1.6	3.2 ± 2.0	1.0 ± 0.6	1.0 ± 0.4
*Melanelia hepatizon* (Ach.) A. Thell	1.2 ± 0.3	1.0 ± 0.2	2.8 ± 2.2	1.3 ± 0.8	1.1 ± 0.3	+	1.6 ± 0.5	1.2 ± 0.4	+	1.4 ± 0.4	1.1 ± 0.3	1.3 ± 0.8	+	+	1.0 ± 0.7	+	+	1.3 ± 0.4
*M. stygia* (L.) Essl.	2.8 ± 1.6			+	+	+	4.0 ± 1.0			2.8 ± 1.6				+				+
*Parmelia omphalodes* (L.) Ach.	+		+		+		+			+		+		+			+	+
*P. saxatilis* (L.) Ach.	+		+		+		+	+		+	+							+
*P. skultii* Hale	1.1 ± 0.4	+	+	1.0 ± 0.4	+	1.5 ± 0.7	+	1.9 ± 1.2	+	1.4 ± 0.6	+	1.0 ± 0.6	+	+	+	+	+	+
*Peltigera aphthosa* (L.) Willd.	+	+	+	1.0 ± 0.6	+		+	1.0 ± 0.5		+	+		1.0 ± 0.6			1.0 ± 0.6		+
*P. canina* (L.) Willd.	+	+	+		+	+	+			+	+			+			+	+
*P. elisabethae* Gyeln.		+			+					+								
*P. leucophlebia* (Nyl.) Gyeln.	+	+		+	+		+	+		+	+	2.0 ± 1.3	+		+	+	+	+
*P. malacea* (Ach.) Funck	+		+		+	+					+			+				+
*P. polydactylon* (Neck.) Hoffm.		+						+		+								
*P. ponojensis* Gyeln.	+	1.0 ± 0.3		+	+	1.0 ± 0.6				+	+		+		+	+	+	+
*P. rufescens* (Weiss) Humb.	+	+	+		+					+	+		+		+		+	+
*Physcia caesia* (Hoffm.) Fürnr.					+									+			+	
*Physconia muscigena* (Ach.) Poelt	+		1.0 ± 0.6		+	+	1.0 ± 0.6			+	+		+	1.0 ± 0.5	+		+	1.0 ± 0.5
*Rusavskia elegans* (Link) S. Y. Kondr. & Kärnefelt			+		+									+			+	
*Solorina bispora* Nyl.		1.5 ± 0.7				2.0 ± 0.9	1.0 ± 0.6				1.5 ± 0.7					2.0 ± 1.4	1.0 ± 0.7	
*S. crocea* (L.) Ach.	+	2.0 ± 1.4			+	+	1.3 ± 0.6			+	1.3 ± 0.7		1.0 ± 0.6			+		1.4 ± 0.6
*S. saccata* (L.) Ach.	1.0 ± 0.7						1.0 ± 0.5				1.0 ± 0,4						1.0 ± 0.7	
*Umbilicaria aprina* Nyl.	+				+		+			+	+					+		
*U. arctica* (Ach.) Nyl.	15.3 ± 10.8				30.0 ± 12.9		+			+	3.0 ± 1.8							12.0 ± 8.9
*U. cylindrica* (L.) Delise ex Duby	15.0 ± 10.3	+			15.0 ± 7.4				+	7.8 ± 5.2	+						+	
*U. decussata* (Vill.) Zahlbr.	1.6 ± 0.8	+	1.3 ± 1.0		1.8 ± 0.7		+		+	1.0 ± 0.6	1.6 ± 0.8			+			+	1.8 ± 0.9
*U. hyperborea* (Ach.) Hoffm.	2.8 ± 1.3	3.7 ± 2.7	1.3 ± 0.8	2.7 ± 1.2	1.1 ± 0.3	+	5.9 ± 2.1		+	3.9 ± 1.5	+			+	1.0 ± 0.6	+	+	1.0 ± 0.6
*U.* cf. *lyngei* Schol.			+		+									+			+	
*U. proboscidea* (L.) Schrad.	1.6 ± 0.8	2.3 ± 1.2		1.2 ± 0.7	+	+	2.8 ± 1.0			2.4 ± 1.2	1.8 ± 1.3		+	+		+		1.3 ± 0.8
*U. torrefacta* (Lightf.) Schrad.	8.5 ± 5.2	1.0 ± 0.5	+		8.0 ± 4.2	+	2.0 ± 0.9				1.3 ± 1.0			8.0 ± 6.9		+	8.0 ± 4.8	2.0 ± 1.6
*U. virginis* Schaer.	+		+		+	+	+			+		+		+			+	+
Number of foliose species	28	21	19	11	30	19	25	11	10	27	26	9	13	20	13	16	23	24
**Fruticose lichens**																		
*Alectoria ochroleuca* (Hoffm.) A. Massal.	2.1 ± 1.1	1.1 ± 0.3	1.0 ± 0.4	1.3 ± 0.8	+	+	2.7 ± 1.4	2.0 ± 1.2		2.6 ± 1.0	+			+		+	+	+
*A. nigricans* (Ach.) Nyl.	2.4 ± 0.5	1.4 ± 0.3	3.1 ± 1.7	2.2 ± 1.7	2.1 ± 0.6	1.2 ± 0.7	2.3 ± 0.6	3.1 ± 1.0	1.4 ± 0.8	2.4 ± 0.5	1.7 ± 0.3	1.0 ± 0.4	+	4.8 ± 3.2	2.5 ± 1.8	1.3 ± 0.6	3.2 ± 1.6	1.5 ± 0.4
*Bryocaulon divergens* (Ach.) Kärnefelt	2.9 ± 0.7	2.2 ± 0.5	3.6 ± 1.3	4.1 ± 2.7	3.1 ± 0.7	1.0 ± 0.4	2.6 ± 0.7	2.5 ± 1.7	1.7 ± 1.0	2.8 ± 1.0	2.7 ± 0.5	1.4 ± 0.7	1.0 ± 0.6	5.0 ± 2.9	2.0 ± 0.7	1.3 ± 0.6	4.3 ± 2.7	2.5 ± 0.6
*Bryoria chalybeiformis* (L.) Brodo & D. Hawksw.	+				+		+	+		+	+	+			+		+	+
*Cetraria aculeata* (Schreb.) Fr.	1.0 ± 0.7	+			2.0 ± 1.2		1.0 ± 0.5		+	2.0 ± 1.2	+				1.0 ± 0.5		+	
*C. muricata* (Ach.) Eckfeldt	1.9 ± 1.0	+	+		1.8 ± 1.1	+	1.8 ± 1.3			1.5 ± 0.8	2.0 ± 1.4			+	5.0 ± 2.6	+	+	
*Cladonia amaurocraea* (Flörke) Schaer.	1.0 ± 0.4				1.0 ± 0.7		1.0 ± 0.4			1.0 ± 0.6	1.0 ± 0.4					1.0 ± 0.5		
*C. borealis* S. Stenroos	+				+		+	+		+	+				+			+
*C. carneola* (Fr.) Fr.		+			+		+				+							+
*C. chlorophaea* (Flörke ex Sommerf.) Spreng.	1.0 ± 0.4		1.0 ± 0.7		1.1 ± 0.3	+					1.2 ± 0.5		+	1.0 ± 0.7				1.0 ± 0.3
*C. coccifera* (L.) Willd.	1.5 ± 0.7	2.5 ± 0.7			1.5 ± 0.5		2.5 ± 0.5				2.3 ± 0.4		1.0 ± 0.7				2.0 ± 1.2	2.0 ± 0.7
*C. gracilis* (L.) Willd.	4.6 ± 3.2	+	+		+		3.5 ± 2.3	+		3.4 ± 2.4	+	+				+	1.0 ± 0.7	+
*C. macroceras* (Delise) Hav.		1.3 ± 0.9			1.3 ± 0.7						+	2.0 ± 0.9						1.3 ± 0.9
*C. phyllophora* Hoffm.	1.0 ± 0.7	+			1.0 ± 0.6	+					+			1.0 ± 0.7		+	1.0 ± 0.6	
*C. pleurota* (Flörke) Schaer.	+	+				+	+			+	+						+	
*C. pocillum* (Ach.) Grognot	+	+			1.0 ± 0.5	+	+			+	1.2 ± 0.8	1.0 ± 0.6	+			+	1.6 ± 0.9	+
*C. pyxidata* (L.) Hoffm.	1.5 ± 0.4	+	+	+	1.0 ± 0.2	+	1.7 ± 0.7	+	1.5 ± 0.8	1.0 ± 0.2	1.4 ± 0.5	1.0 ± 0.6	+	+	4.3 ± 2.3	+	+	1.1 ± 0.3
*C. stricta* (Nyl.) Nyl.	1.9 ± 1.5	+			+		+	6.0 ± 3.8		1.0 ± 0.6	1.3 ± 0.7		+				+	1.3 ± 1.0
*Pseudephebe minuscula* (Nyl. ex Arnold) Brodo & D. Hawksw.	17.7 ± 12.4	+			25.0 ± 11.8	+	1.8 ± 1.3		+	2.8 ± 1.6	+			12.8 ± 9.9	+		12.0 ± 6.2	
*P. pubescens* (L.) M. Choisy	1.8 ± 0.6	+	2.8 ± 1.6		1.3 ± 0.5	+	1.9 ± 0.7	2.8 ± 1.7	+	2.0 ± 0.6	1.0 ± 0.6			+	+	+	+	1.6 ± 1.1
*Sphaerophorus fragilis* (L.) Pers.	5.7 ± 3.2	1.0 ± 0.6	+	+	6.6 ± 3.2	+	3.4 ± 2.0	2.0 ± 1.2	+	5.7 ± 3.2	+	+	+	1.3 ± 1.0		+	1.3 ± 0.9	+
*S. globosus* (Huds.) Vain.	2.9 ± 0.8	2.8 ± 1.5	+		2.7 ± 1.4	+	3.9 ± 0.8	1.3 ± 1.1	+	3.1 ± 0.9	1.7 ± 0.7			+	+	2.2 ± 1.7	+	2.0 ± 1.2
*Stereocaulon alpinum* Laurer	2.4 ± 0.6	5.3 ± 2.7	+	2.0 ± 1.3	1.2 ± 0.8	1.9 ± 0.8	2.1 ± 0.5	7.5 ± 4.9	11.5 ± 2.7	3.7 ± 2.5	2.7 ± 0.7	1.8 ± 1.3	11.0 ± 7.7	1.6 ± 0.9	4.0 ± 2.5	7.9 ± 5.2	2.1 ± 1.0	3.5 ± 1.0
*S. botryosum* Ach.	3.1 ± 1.6	4.8 ± 1.8	5.0 ± 2.7		2.5 ± 1.3	4.6 ± 3.5	8.0 ± 3.9	2.0 ± 1.3	5.0 ± 1.3	6.0 ± 2.2	3.5 ± 1.9	2.5 ± 0.7			1.4 ± 0.7	1.2 ± 0.8	2.0 ± 1.2	3.5 ± 2.5
*S. condensatum* Hoffm.	2.0 ± 0.8	1.0 ± 0.4					2.0 ± 1.4	1.0 ± 0.8	1.0 ± 0.7	1.5 ± 0.3								
*S. depressum* (Frey) I. M. Lamb		1.0 ± 0.3					1.0 ± 0.4	1.0 ± 0.6		1.0 ± 0.6				1.0 ± 0.8			1.0 ± 0.3	
*S. glareosum* (Savicz) H. Magn.	2.3 ± 1.1				4.0 ± 2.3				+	2.3 ± 1.5								
*Thamnolia vermicularis* (Sw.) Schaer.	2.0 ± 0.3	2.0 ± 0.3	2.3 ± 0.7	3.2 ± 1.8	1.9 ± 0.3	1.3 ± 0.3	2.6 ± 0.5	2.8 ± 1.0	1.5 ± 0.6	2.3 ± 0.4	2.0 ± 0.3	2.1 ± 0.6	1.4 ± 0.8	1.4 ± 0.6	2.5 ± 1.4	1.8 ± 0.5	1.6 ± 0.4	2.1 ± 0.3
*Usnea sphacelata* R. Br.	+	+	+		+		+		+	+	+	+	+		+			+
Number of fruticose species	26	24	14	7	26	16	25	16	14	24	26	12	11	14	14	15	21	20

Note. +—presence of a lichen with cover <1%.

**Table 3 plants-13-00193-t003:** Eigenvalues and calculation of the variance of the components.

Components	Eigenvalue	% of Total Variance	Cumulative Eigenvalue	Cumulative, %
Crustose lichens				
1	6.8074	13.0911	6.8074	13.0911
2	4.0709	7.8286	10.8782	20.9197
3	3.0108	5.7901	13.8891	26.7098
4	2.8326	5.4473	16.7216	32.1570
5	2.6450	5.0861	19.3667	37.2436
6	2.3047	4.4321	21.6714	41.6757
7	2.0824	4.0045	23.7537	45.6802
8	1.9932	3.8331	25.7470	49.5134
9	1.6285	3.1317	27.3755	52.6451
10	1.5972	3.0714	28.9726	55.7166
Fruticose lichens				
1	3.5761	12.7720	3.5761	12.7720
2	2.3682	8.4578	5.9443	21.2298
3	1.9041	6.8004	7.8484	28.0302
4	1.5328	5.4741	9.3812	33.5043
5	1.4521	5.1860	10.8333	38.6903
6	1.3175	4.7052	12.1508	43.3956
7	1.2920	4.6142	13.4427	48.0097
8	1.1600	4.1427	14.6027	52.1525
9	1.1200	4.0001	15.7227	56.1524
10	1.0510	3.7537	16.7737	59.9062

**Table 4 plants-13-00193-t004:** Results of the multiple regression analysis for orographic factors and components (n = 130).

Orographic Factor	b*	s*	b	s	*t*	*p*
Crustose lichens (*R*^2^ = 0.3425)						
Altitude	0.5691	0.1489	0.0399	0.0127	3.1514	0.0024
Distance to the glacier	0.4655	0.1233	0.1285	0.0381	3.3696	0.0013
Slope angle	−0.3766	0.1508	−0.0395	0.0158	−2.4976	0.0150
Foliose lichens (*R*^2^ = 0.2218)						
Slope angle	−0.4323	0.1421	−0.0229	0.0085	−2.6911	0.0089
Fruticose lichens (*R*^2^ = 0.3876)						
Exposure	0.4974	0.1297	0.1974	0.0678	2.9109	0.0048
Slope angle	0.4131	0.1442	0.0256	0.0118	2.1709	0.0333
Altitude	−0.3990	0.1423	−0.0228	0.0109	−2.1009	0.0392

Note. b*, s*—standardized regression coefficients and their errors; b, s—nonstandardized regression coefficients and their errors; *t*—Student’s criterion; *p*—significance level; *R*^2^—coefficient of multiple determination between all variables and response (2nd degree polynomial); n is the number of test areas (relevés) on the basis of which the model is built. The variables in the table are arranged in descending order of the coefficient b*.

## Data Availability

The specimens of lichens are stored in the herbarium of the Komarov Botanical Institute of the Russian Academy of Sciences (LE). No experiments were conducted on humans or animals. The programs and methods available and widely used in the works on our subject were used, which can be found in the published works [28,29,30,31,32,33,34,35,36,37,38,39].

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
