# Peer review of "Influence of Orographic Factors on the Distribution of Lichens in the Franz Josef Land Archipelago"

_plants, 2024, doi:10.3390/plants13020193_

Round 1

Reviewer 1 Report

Comments and Suggestions for Authors

Lots of good information in the Introduction but early on you need to State the general location and climate of the study area. Otherwise much of the arctic weather discussion is irrelevant.

"Retreating, glaciers currently leave a hilly ridge 130 topography with absolute heights not exceeding 70–80 m". A photo of this land type would be helpful.

Results:

I did not see a Methods section? So I have no idea what the plot size is. There can not be a total average lichen cover without some parameters? Size of plot, per boulder, or some parameters? This makes commenting on the results impossible. I am impressed with Table one identification of so many challenging lichen determinations but the quantitative comments can not be evaluated without a plot? One could give relative abundance ratings such as low, medium and high but those need to be defined similar to how forest health plots are conducted? 

Table. 1. 

A good way to show which taxa were considered by each growth form since some taxa can be placed in more than one growth form.

please define if the symbol + means trace amount or present versus absent? 

The numbers are difficult to evaluate when they are all so small in average cover?

But average total lichen cover by species does not really quantify the data. This needs to be quantified to communicate to other scientists. I am sure the authors know what they mean by this but it needs to be communicated.

Figure 1. would be better if the scale for total cover remained the same for each variable. So the reader has a better visual relative abundance of each variable.

How can the total cover value be 80 for a growth form of lichen when we lack a plot?

and is it 80% of the plot or what is 80?

OK, I found the "materials and Methods on page ten after the results and after the Discussion???

This needs to be fixed. Also rating % total cover by growth form in the field and then reporting % total cover in a large 4 x 4 meter plot later after keying the individuals out later by species and by a different personnel!  Is impossible.  One can not divide up a complex mosaic of crustose species a few months or a year later by species to % cover. There are too many look alike taxa and the same taxa that might be vegetative or fertile in the field and look different. I doubt that each species in each plot was collected and verified and vouchered? Maybe they were vouchered and if so state that. However, in the field it is difficult to distinguish even the generic level. The Rinodina species in the field look similar to Lecanora species and one needs to see the color of the spores to be sure of the taxa. Also I am not clear how the total percent cover was actually measured in these 4 x4 meter plots. It sounds like you would just rate the ocular estimate for the ecological micro- and nanorelief areas within the 4 x4 plot? if so state that but it is misleading to see these high percent cover values for these large plots?  photos of one or two plots would help.

line 544-546

"Cover was visually assessed for all species or groups of lichens noted in the field. Identification of lichen collection was carried out by the second and third authors using morphological, anatomical and chemotaxonomic methods "

Chemical methods were done in the lab so knowing all the taxa in the field and rating each taxa cover value is not possible.

line 565

"The average value of cover was calculated for each lichen species within the specified ranges" Does this mean each species including unknown field taxa and for every gradient and every plot and cover for the entire 4 x 4 meter plot or only for the focused micro- environment such as a crack in the rock?

line 570-571

" When calculated the cover for blocks, flagstones and boulders, we took into account only the cover of lichens located on faces parallel to the general slope and general orientation of the site." This is a confusing statement please make it clear how you rated the cover value. I realize rating a boulder which is round is difficult compared to the classic flat plot. I suspect your cover values where focused only on the one slope of the rounded or angular boulder but this needs more clarification. I also believe that your cover values are "relatively" rated based on some more focused parameters than the entire 4 x 4 meter plot. Did you have a tape measure for this plots or how did you lay it out on such irregular and steep slopes? the reader should be able to go out and replicate your methods.

Line 426-428

"bush lichens increases at altitudes exceeding 60 m above sea level. This increase is attributed to the prevalence of flagstone at these altitudes, which crevices facilitate the settlement of lichen patches, most often comprising several species." Define bush lichens are they a subset of fruticose lichens or are they all the fruticose lichens?

You tell us where the specimens collected are stored but you do not make it clear how many specimens were collected. All taxa, one specimen of each one species? from each plot? only one representative of each macolichen and every crustose lichen in each plot? This is a difficult and time consuming task. So some easy to determine specimens might not warrant collection from each plot? Tell us how this was done.

Were any duplicates collected? will duplicates be sent to other herbaria?

4 x 4 meter plot are very large plots to ocularly estimate percent cover. 

Plot selection is bias and needs some criteria for selection even if select is stratified random? Stratified plots are fine but there needs to be criteria to select them? If not it is not science.

Line 565

"The average value of cover was calculated for  each lichen species within the specified ranges". 

Is this average value for the total of all the plots or for each individual plot? This seems to be difficult to conduct and confidence interval is most reported as plus or minus 0 in Table 1.

line 570-571

"When calculated the cover for blocks, flagstones and boulders, we took into account only the cover of lichens located on faces parallel to the general slope and general orientation of the site" so this is not for the entire 4x4 meter plot? this needs to be noted in the figures.

Figure 4. Tell us more about the map. What are the green areas on the islands? Ice free areas? 

Comments on the Quality of English Language

Most of the paper is well done. The discussion is a bit awkward and could be improved. The Methods are too brief and confusing but this may not be due to the English.

Author Response

Thank you very much for taking the time to review this manuscript. Please find the detailed responses in attachment below and the corresponding corrections highlighted color in the re-submitted files.

Reviewer 2 Report

Comments and Suggestions for Authors

First the good news: the actual research is on a good topic and is a very good investigation into the lichens of a relatively unknown area. In my view, this means that publication is worthwhile.

However, the ot-so-good news, the overall writing style and organisation of the paper is poor and can be greatly improved.

I would suggest that a 20% cut in length is achievable by a better structure.

Introduction:

I suggest that this can be restructured to better introduce the paper and lead into the actual research.

The real information starts at line 74 and I suggest:

General introduction, what the topic is and why it is important and some examples of similar work plus gaps in our knowledge. The authors seem not to know about southern hemisphere succession as glaciers retreat. This is especially interesting in Chile where the presence of early very active nitrogen fixers results in rapid succession, bare ground to mature forest in 30 years, lichens growing at around 8 mm per year and colonising bare rocks in months rather than decades. I do agree that succession is much slower in the Arctic, probably due to the lack of a good N-fixer.

This section could end with a set of key research questions that are being considered in this paper. The advantage of this is that the Discussion sort of writes itself.

The next sections can describe the research area.  Information in this section should not be repeated in the methods.

Figures:

My view is that a reader should be able to understand a Figure without having to read the full caption.

Figures in this paper need to have their axes labelled on the graphs; 

Captions should be slightly changed so that the label for a panel (a, b, c etc) leads the information about the panel, here the label follows the description and this in uncomfortable to readers.

In some Figures a comma is used instead of a point for the decimal (I know this is normal is some countries but not usually in published papers in English language journals).

In Fig. 2, caption, what does absolute height mean, I suggest to consistently use altitude.

Table, this is a large Table with a lot of information. The authors could consider putting it in the Supplementary section but, possibly, the journal editors will see little problem in its presentation in the main paper.

In the Table, what does a + mean?

In the column labels, I think Latitude should be altitude (looks like a spell checker automatic change).

In the caption give the units for slope (degrees?) and also define cover (this can be % or sometimes a straight count), so clarification needed.

In the text, please give the full units for temperature °C and not just ° as this can be confused with slope.

Line 379, what is a cap lichen.

Line 551, what does aspect by country of the world mean?

In the PCA analysis can the authors use site characteristics as well? Eg slope, height, aspect, and see where they lie on the PCA axes.

Problem: the authors seem to have no ages for the sample sites in front of the glaciers. They use distance as a proxy but this is unreliable if the glaciers retreat at different rates or in bursts.

Is there any chance of getting ages?

Also, although used in the discussion about the lichens the authors also have no good environmental data for the climate, such as actual temperatures, change with altitude, effect of cloudiness (fog water deposition)

The apparent lack of this information means that I do not support some of the suggestion made here about links between lichen occurrence and factors such as temperature.

The authors should make an attempt to present the results and discussion in a more organise way, eg dealing with lichen forms separately, looking at general cover separately. At the moment it is very difficult to read.

Another problem is that the data are from diverse sites across the archipelago and this make interpretations more difficult compared to a structured sampling in front of a single glacier.

There are also a lot of minor but important points that can be improved.

I assume that the authors do not have English as a first language. The actual writing is good but has problems that are often found in second language writing. First, the overall sentence structure is often too long and many sentences can be shortened or combined.  Second, the authors use some words that are, as far as I know, technically perfectly correct but are not words that a are typically use by a native speaker.

Example 1, the use of monotonous when describing the shape of a graph.

Eg: All lichen groups are characterized by a monotonous increase in the number of species.

The authors have fitted curves to the data and it is better to use the form of the fit as a description of the curve form, eg linear, parabolic, exponential.  Not only are these more understandable to most readers but are also better descriptors of the fitted line. I had a look on the web and there seem to be several pages which give and define words that can be used to describe a graph.  These could be useful.

Example 2, use of rarely used or highly specific words, a good example is on line 437, hypsometrically higher; as far as I can tell this means height above sea level. Altitude would be a better word.

Comments on the Quality of English Language

I dislike commenting on the quality of the English because I know myself how hard it is to write at a good level in a second language. The problem in this paper is that words are often used that are technically correct but would be rarely or never used by a native speaker. I have commented on these in the report. A check by native speaker would certianly help but a rewrite to meet my comments is also important.

Author Response

(The authors gave the same response as above.)

Reviewer 3 Report

Comments and Suggestions for Authors

The authors presented results of study about lichen species distribution along the graditents of several orographic factors in the Franz Joseph Archipelago. The valuable achievement is analyses of various ways of lichen colonization as the results of glacier melting, which is the urgent global climate change issue. The relevant methodology had been applied to determine lichen species, estimate their cover, and large sets of numerical data had been obtained. The authors used relevant statistics methods for presenting results, and for assessing impact of selected orographic factors on lichen diversity and distrubution.

Some specific comments and suggestions are given below, as it follows:
Row 2-3: the paper title, consider to change it, include "distribution" because in the abstract you indicated "the significance of orographic factors in the distribution of lichens was assessed..." This can be an idea to change the title in order to better reflect the main topic..
Row 16-17: "Along the gradient of altitude above sea level", suggestion to change: "Along the altitudinal gradient"
Row 126: subchapter 1.1.1. Relief is very long and gives too much descriptions which is more suitable for textbook. Try to descibe the moste relevant relief features of the area. 
Row 198-199: delete "85% of this amount to solid precipitation", you repeated it unintentionally.
Row 406: chapter Discussion, Table 4 is too long, is it really necessary ? Data in Table 4 are related to description of the relief forms, so it does not require to be included in chapter Discussion.

Comments on the Quality of English Language

The quality of english language is good.

Author Response

(The authors gave the same response as above.)

Round 2

Reviewer 1 Report

Comments and Suggestions for Authors

()

Comments on the Quality of English Language

()

Author Response

Thank you very much for taking the time to review this manuscript.
The linguistic revision of the manuscript was carried out by Prof. Mark Seaward (University of Bradford, United Kingdom). The corresponding corrections highlighted color in the re-submitted files. 
In the Acknowledgments, we thank the anonymous reviewers for their valuable comments that improved the manuscript. Prof. Mark Seaward declined to acknowledge the thanks.

Reviewer 2 Report

Comments and Suggestions for Authors

The authors have made several changes following the first review and are thanked for doing this. 

Some minor chages are suggested (ie: change what is written to)

L 171   with increase in distance . . . 

L 400  avoid higher heat input and consequentioal more rapid moisture . . .

L 515   were allocated to three . . . 

L 592 change hypsometric to above sea level

Comments on the Quality of English Language

They did ask about the English and I am happy to leave this as submitted as there is no great difficulty in understanding what is written.

Author Response

Thank you very much for taking the time to review this manuscript. We have accepted your corrections.
The linguistic revision of the manuscript was carried out by Prof. Mark Seaward (University of Bradford, United Kingdom). The corresponding corrections highlighted color in the re-submitted files. 
In the Acknowledgments, we thank the anonymous reviewers for their valuable comments that improved the manuscript. Prof. Mark Seaward declined to acknowledge the thanks.

Reviewer 3 Report

Comments and Suggestions for Authors

After the first reviewing process of the submitted paper, the authors seriously considered suggestions and recommendations. The second version of the paper is significantly improved, more comprehensive and extended. The content of the paper is now very understandable. 

As a Reviewer I am confirming that paper is acceptable to be published in journal.

Comments on the Quality of English Language

The English language is proper.

Author Response

(The authors gave the same response as above.)
